# Costs and models used in the economic analysis of Total Knee Replacement (TKR): A systematic review

Naline Gandhi[1]☯, Amatullah Sana Qadeer[1]☯, Ananda Meher[1]☯, Jennifer Rachel[1], Abhilash Patra[1], Jebamalar John[1], Aiswarya Anilkumar[1], Ambarish Dutta[2], Lipika Nanda[1], Sarit Kumar Rout[2]*

**1** Indian Institute of Public Health-Hyderabad, Public Health Foundation of India, New Delhi, India, **2** Indian Institute of Public Health-Bhubaneswar, Public Health Foundation of India, New Delhi, India

☯ These authors contributed equally to this work.
* sarit.kumar@phfi.org

## Abstract

### Objectives

The main objective of this review is to summarize the evidence on the core modelling specifications and methodology on the cost-effectiveness of TKR compared to non-surgical management. Another objective of this study is to synthesize evidence of TKR cost and compare it across countries using purchasing power parity (PPP).

### Methodology

The electronic databases used for this review were MEDLINE (PubMed), Cochrane Central Register of Controlled Trials (CENTRAL), HTAIn repository, Cost effectiveness Analysis (CEA) registry, and Google Scholar. Consolidated Health Economic Evaluation Reporting Standards (CHEERS) was used to assess the validity of the methods and transparency in reporting the results. The Quality of Health Economic Studies (QHES) was used to check the quality of economic evaluation models of the studies included. The cost of TKR surgery from high income and low- or middle-income countries were extracted and converted to single USD ($) using purchasing power parities (PPP) method.

### Result

Thirty-two studies were included in this review, out of which eight studies used Markov model, five used regression model, one each reported Marginal structure model, discrete simulation model, decision tree and Osteoarthritis Policy Model (OAPol) respectively to assess the cost-effectiveness of TKR. For PPP, twenty-six studies were included in the analysis of TKR cost. The average cost of TKR surgery was the lowest in developing country—India ($3457) and highest in USA ($19568).

**Data Availability Statement:** All relevant data are within the manuscript and its Supporting Information files.

**Funding:** This study was funded by Department of Health Research, Ministry of Health and Family Welfare, Government of India to Regional Resource Hub-HTAIn, Indian Institute of Public Health, Hyderabad. The funders had no role in study design, data collection and analysis, decision to publish, or preparation of the manuscript.

**Competing interests:** The authors have declared that no competing interests exist.

## Conclusion

The findings of this review showed that the Markov model was most widely used in the analysis of the cost effectiveness of TKR. Our review also concluded that the cost of TKR was higher in the developed countries as compared to the developing countries.

## Introduction

Osteoarthritis (OA) is a degenerative joint disease involving the cartilage and surrounding tissues and is the leading cause of disability worldwide among older adults [1]. Globally, the prevalence of OA knee was estimated to be around 22.9% [2] whereas in India it was estimated to be 28.7% [3]. The direct cost of OA knee was US$5294 per person per year for those aged over 65 years and $5704 for patients less than 65 years. This was estimated to be twice of the non-OA patients. The Indirect costs of OA knee was around US$4603 per person per year, mainly due to work-related losses and home-care costs. The Total knee replacement (TKR) is considered as one of the interventions to overcome the burden of OA knee. The number of TKRs being done to mitigate the burden of OA knee has also been increasing throughout the world. In the United States of America, approximately 700,000 TKR surgeries were performed in the year 2010, and its demand is predicted to grow to 3.8 million per annum by the year 2030 [4].

The cost of TKR in developed countries—USA is around $17500 (in 2017) and in the European countries—Denmark and UK are €13149 (in 2020) and £7313 (in 2013) respectively, however, in the developing countries like India, it is ₹80,000 (in 2021) [5–8]. Many developed countries have considered cost-effectiveness analysis as one of the methods for policy level decision-making. However, India, which has multiple health system constraints and limited government investment on health, is progressively preparing to include cost-effective analysis as a tool for decision-making at the policy level. There has been only one study conducted in India, which showed that the TKR is cost-effective in the base case scenario with an Incremental Cost Effectiveness Ratio (ICER) of ₹9789 ($132.3) per QALY [7].

Given the emerging disease burden of osteoarthritis in both developed and developing countries and its associated interventions involving huge financial burden both to the individuals and government, it is critical to review the current literature regarding the cost effectiveness of such interventions and the impact of the prevailing cost on the health systems. Moreover, given that varieties of models- the decision tree, Markov model, and regression analysis [9] were used in these types of studies, identifying a suitable model also assumes significance. During our literature search, four systematic reviews were identified which compared the cost-effectiveness of TKR to non-surgical management in patients with OA knee. Two reviews compared the cost-effectiveness of TKR using ICER value irrespective of the models used [10,11]. A recent review by Kamaraj et al [12] assessed the scope and quality of all current cost-effectiveness analysis (CEA) studies for TKR in order to identify trends, and recognise the areas for improvements. Further, another review by Lan et al [13] focused on the study design and compared the Markov model with RCT and tree diagram in the analysis of cost-effectiveness of TKR.

However, none of the systematic reviews mentioned above suggested a suitable method for the analysis of cost-effectiveness of TKR. Therefore, we aim to perform a systematic review to understand the methods used in economic evaluations/cost-effectiveness of TKR compared to non-surgical management of OA knee. The main objective of this study is to summarize the evidence on the core modelling specifications and methodology on the cost-effectiveness of

TKR compared to non-surgical management. Another objective is to synthesize evidence of cost of TKR and make cost of different countries comparable using purchasing power parity (PPP) method, which has not been done before.

## Methodology

A systematic review protocol was registered in International Platform of Registered Systematic Review and Meta-Analysis Protocols (INPLASY) with registration no: (INPLASY202290044). Cochrane methodology was adopted and the Preferred Reporting Items for Systematic Review and Meta-analysis (PRISMA) guidelines were used for the purpose of reporting.

### Criteria for considering studies in the review

**Types of studies.** In this review, all the reports of randomized control trials (RCT's) and cohort studies were included. Moreover, cross-sectional, observational, and case-control studies that can provide information on cost-effectiveness, cost benefit analysis, and cost utility analysis were also included in our initial search.

**Types of participants.** Participants aged 40 years and above who have primary OA knee.

**Type of intervention.** Participants with OA knee who underwent surgical intervention that is total knee replacement (TKR).

**Comparator.** Participants with OA knee without surgical intervention (TKR) or underwent non-surgical management.

**Types of outcome measures.** Economic evaluation studies that report outcomes—Incremental Cost-effectiveness Ratio (ICER), cost-effectiveness, and improvement in QALY.

### Search strategy for identification of the studies

The electronic databases included in the search were MEDLINE (PubMed), Cochrane Central Register of Controlled Trials (CENTRAL), HTAIn repository CEA (Cost-effective analysis) registry, and Google Scholar. The search history was conducted from the earliest possible date till 18th November 2022 and filters were not applied for time period. The computer-based search terms included the combination of keywords and Mesh terms like "Total Knee Arthroplasty" and "Cost-Effectiveness" 'Knee osteoarthritis', 'Cost-utility analysis', 'Total Knee Replacement' and 'Economic Evaluation' 'Non-surgical Management" and comparators (NSAIDs, steroids, visco-supplementation, physiotherapy, exercise), and outcome (quality of life, EQ5D, ICER). Snowballing technique was also performed to identify other relevant articles. The search strategies for the electronic database are in the S1 Appendix.

### Inclusion and exclusion criteria

1. Studies with English language were included

2. Studies with participants aged 40 years and above with degenerative OA knee were only included whereas traumatic OA knee-based studies were excluded.

3. Study design such as Randomized control trial or quasi-randomized control trial-based on TKR (+ Postsurgical management) study and/or non-surgical management with cost-effectiveness, cost benefit analysis, cost utility analysis along with prospective observational study on TKR (Post-surgical Management) and/ or non-surgical management were included.

4. Studies that report costing, Quality- Adjusted Life Years (QALYs), and economic models for calculating ICER of TKR, TKR (+post-surgical management), and TKR vs non-surgical management were included.

5. Studies considering other surgical procedures- Partial Knee Arthroplasty (PKA), Unicompartmental Knee Arthroplasty (UKA), and Kinespring Implant were excluded. Simultaneous bilateral TKR and revision TKR studies were also not included.

6. Studies were excluded if this involved incomplete economic analysis: only cost comparison or only QALY comparison were reported. Non-economic evaluations were also excluded.

7. Studies—systematic reviews, letters to the editors, commentaries, and protocols were excluded.

## Selection of studies

All the identified studies from the database were imported to the RAYYAN software and duplicates were removed. During the "first pass", three authors independently reviewed the study titles and abstracts before concluding whether the study should be "included," "excluded," or there is "uncertainty" about it. The consensus was reached in studies that had "uncertainty." The remaining studies were excluded, while those that were "included" were considered for the next stage of selection.

Three authors retrieved the full texts of the selected abstracts from the first stage and screened them again. Similar screening methods were applied, but the authors agreed to include papers through an iterative consultation process for those with conflicting findings. Papers with unresolved conflicts were excluded after this process.

## Data extraction and management

Data from the selected full text articles were extracted for subsequent synthesis. A data extraction framework was developed for extracting data based on—author's name, year, study location, type of model, perspective, direct and indirect cost of TKR and non-surgical management, ICER, and QALYs. Data was extracted by two authors from the selected full-text papers, and it was verified by another two authors. The fifth reviewer was consulted for expert opinion if there was any disagreement during the data synthesis process.

## Conversion of Costing data on Purchasing Power Parities (PPP)

The cost of TKR differs across different metrics—time and currency. Change in the prices due to inflation affects the cost of TKR. This study adjusted the cost of each country to the year 2021, using price index (inflation index) of different countries. Further, the cost from the high income and the LMIC countries were converted to single USD ($) using purchasing power parity (PPP) method (Cost in USD = cost in any currency / exchange rate in PPP). The PPP are the rates of currency conversion that equalize the power of different currencies [14].

## Quality assessment of studies

A 24-item tool was used to assess the validity of the methods and transparency in reporting the results of the study based on the Consolidated Health Economic Evaluation Reporting Standards (CHEERS) [15]. The checklist was designed to qualitatively evaluate an economic study in terms of title and abstract, introduction, methods, results, discussion, disclosure of funding source, and conflict of interest. Two authors evaluated the studies based on these parameters

from the checklist and further, it was verified by another two authors. Expert opinion was drawn from a fifth author wherever necessary. Each item of the study was assigned "Yes" if the information is reported completely; "Part" if partially reported and "No" if not reported. NA was reported if the item was "not applicable". A score of 1 was assigned if the criteria was "Yes", 0.5 if it was "Part" and 0 if the criteria was "No". Hence, the maximum and minimum score for a study were 24 and 0 respectively. Additionally, the percentage was determined under the presumption that each study received equal weighting, with "Not Applicable" item being left out. Studies with a score of at least 75% were defined as high quality, those between 60 to 75% as moderate quality, and those below 60% as low quality.

The Quality of Health Economic Studies (QHES) instrument developed by Chiou et al is a measure to check the quality of economic evaluation models used in the research papers [16]. The instrument contains 16 dichotomous (yes/no) items, where, each item carries a weighted point value generated through data gathered from a survey conducted among an expert international panel of health economists. It uses visual analog scale which has both content and construct validity. The QHES is scored on a 0 (lowest quality) to 100 (highest quality). The quality score is calculated by subtracting points for questions answered with no from 100. Therefore, the highest possible score is 100, and the lowest is 0. Studies with a score exceeding 70 points are considered of high quality. This continuous scale is again grouped by the following categories— High quality (Above 70–100), Fair quality (50–70), and Poor quality (Below 50).

Moreover, the instrument gives a single score which can be compared directly. The CHEERS instrument was used to check the qualitative measures used in the papers whereas QHES was determined to be particularly useful for this systematic review because it is used to evaluate the economic models: cost-effectiveness, cost utility and cost minimization. The QHES tool is a valid tool for measuring cost-effectiveness studies.

The quality assessment did not influence the inclusion or the exclusion of studies. Studies meeting the inclusion criteria were included for the purpose of this review irrespective of the quality criteria.

## Data analysis

Due to heterogeneity of the included studies and reported cost data, formal meta-analysis could not be performed. Thus, each study was analysed qualitatively and primarily presented in tables and graphs.

Out of the thirty-two studies, six studies were excluded from the PPP analysis. It was observed that three were excluded due to non-availability of cost data [17–19] and three due to being published before the year 2000 [20–22], where the data were older than 20 years. Studies conducted before the year 2000 were excluded from the PPP analysis due to change in cost as the chance of technology upgradation could be higher over a period of time. The remaining twenty-six studies [5–8,23–44] that were published after the year 2000 and presented direct cost of TKR were included in our analysis. The TKR cost of all studies was inflation adjusted to 2021 price in their respective country's currency. The country wise inflation rate (in term of consumer price index) and the PPP value were extracted from the world development indictors data base (published by the World Bank) [44].

## Results

The search yielded total 820 articles (784 articles from the PubMed database, 25 articles from Cochrane central, 2 studies from CEA registry database and 9 from Google scholar). After removing duplicates (30 articles), 790 articles were eligible for title and abstract screening. After initial screening, 79 articles were eligible for full text screening. Out of 79, full text was

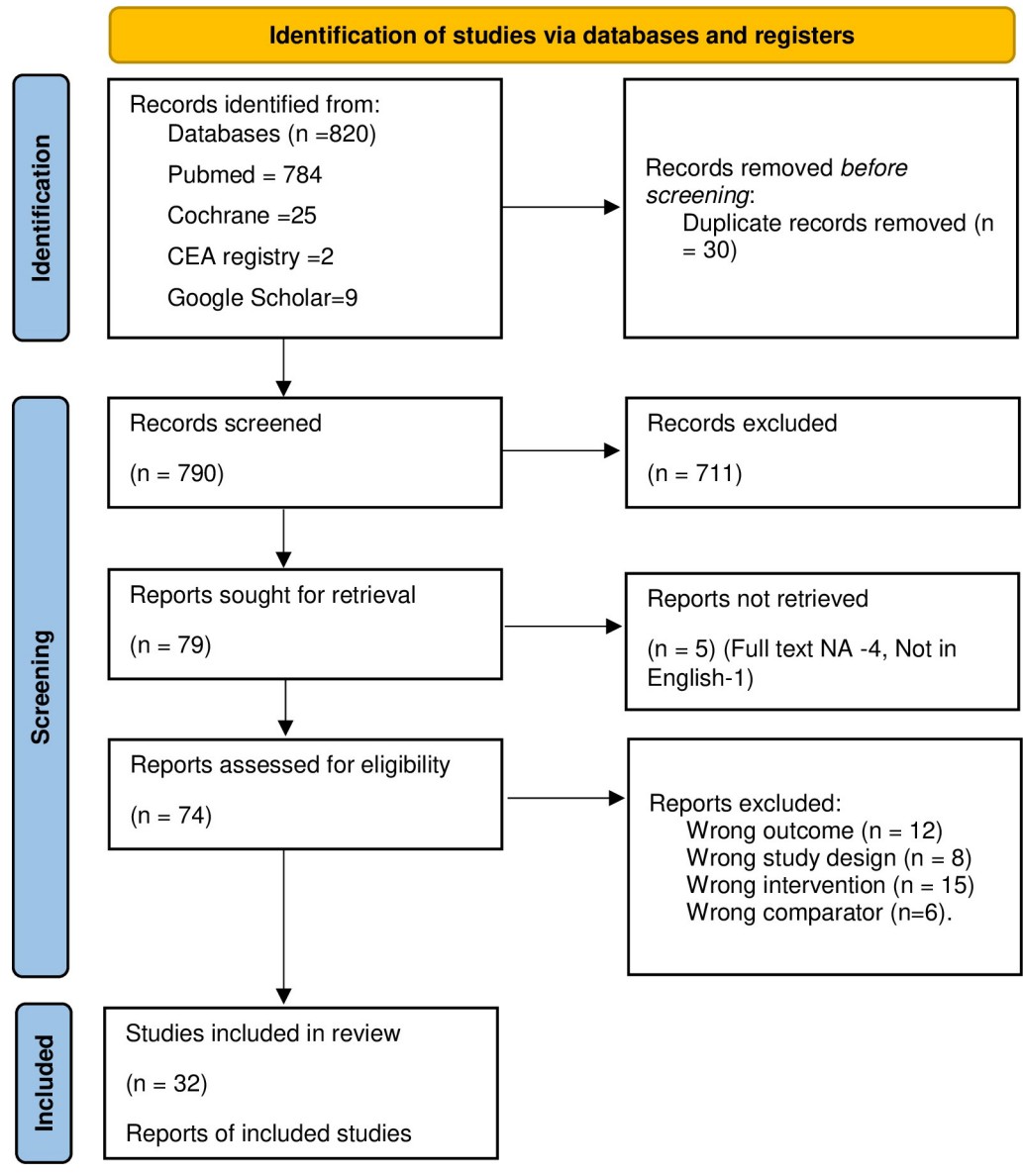

**Fig 1. Studies selection process through PRISMA.**

not available for 4 articles and 1 was not in English language, hence were excluded. Among the retrieved 74 full text articles, 42 articles did not meet the inclusion criteria. The reasons for exclusion were: articles with different outcomes (12 articles), other study design (8 articles), intervention other than TKR (15 articles) and comparators other than non-surgical management like UKA, PKA (7 articles). Hence, 32 studies were included in this review for the descriptive synthesis. The process of selection of studies are mentioned in PRISMA flowchart (Fig 1).

## Study characteristics

**Description of included studies.** Thirty-two studies [5–8,17–43,45] met the inclusion criteria and were published between 1993 and 2022. The included studies were from different

**Table 1. Characteristics of the selected papers.**

| Characteristics | Number of Papers N (%) |
| --- | --- |
| *Year* | |
| Before1995 | 1 (3.1) |
| 1995–2005 | 2 (6.2) |
| 2006–2015 | 14 (43.7) |
| 2016- Current | 15 (46.8) |
| *Europe* | 10 (31.2) |
| *North America* | 16 (49.9) |
| *Asia* | 3 (9.4) |
| *Australia* | 3 (9.4) |
| *Comparators* | |
| Surgery vs non-surgery | 29[1](90.6)) |
| Only surgical approach | 6 (18.7) |
| *Time Horizon* [2] | |
| Lifetime | 13 (40.6) |
| <12 Months | 3 (9.4) |
| 12–24 months | 10 (31.2) |
| >24 months and less than lifetime | 8 (25.0) |
| Not reported | 1 (3.1) |
| *Utility Assessment* | |
| Direct Assessment | 1 (3.1) |
| Analytical Model | 28 (87.5) |
| Not Defined | 3 (9.4) |
| *Cost Assessment* | |
| Only direct Cost | 24 (75.0) |
| Both direct and indirect cost | 7 (21.9) |
| Not reported | 1 (3.1) |
| *Sensitivity Analysis* | |
| Deterministic | 3 (9.4) |
| Probabilistic | 6 (18.7) |
| Not Defined | 9 (28.1) |
| Not Applicable | 14 (43.7) |
| *Discount Rates* | |
| Explicitly defined | 22 (68.7) |
| Not reported | 10 (31.2) |

[1] The studies used more than one comparator and approach. Hence there is an overlapping.

[2] During the follow-up period, the retrieved articles used more than one time horizon to estimate their functional improvement and cost-effectiveness.

parts of the world with twelve studies from United States [5,18,19,22–27,33,41,45] four studies from Canada [21,28–30], three studies from United Kingdom [8,34,35] and two each from Finland [17,20] and Australia [31,32]. One study each was conducted in India [7], China [40], Germany [37], Spain [39], Romania [38], France [36], Denmark [6], New Zealand [43] and Kazakhstan [42] (See Table 1).

Different types of methodologies were used to assess the cost-effectiveness of TKR in the included studies. Decision analytic model that is Markov model were used by nine studies [5,7,18,23,26,28,36,43,45] whereas five studies reported regression model [6,32,33,35,37]. One study each reported a marginal structure model [27], discrete simulation model [31], decision tree analysis [8] and Osteoarthritis Policy Model (OAPol) [41] to assess the cost-effectiveness of TKR. Fourteen studies [17,19–22,24,25,29,30,34,38–40,42] did not report any methodology to determine cost-effectiveness.

Thirteen Studies [7,8,19,23,26,27,30,31,36–38,41,45] considered time horizon to be life time whereas eighteen studies used ≥12 months and less than lifetime of time horizon

[5,5,6,17,18,21,22,28–30,32–36,40,42,43]. Three studies [24,25,39] evaluated time horizon less than a year and one study [20] did not specify time horizon(Table 1).

Twenty-two studies reported discount rates, out of which, thirteen studies [6,7,19,23,26–28,30,35,36,38,39,41–43,45] discounted cost and QALY at 2% to 6%, of which, four studies [5,18,41,42] discounted only cost at 3% and three studies [8,17,43] had discounted only QALY at 3.5% to 5%, while another study [31] discounted cost and Disability Adjusted Life Years (DALY) at 3%. Only one study [37] discounted benefit at 3%. Two studies [25,34] did not apply discount rates and eight studies [20–22,24,29,32,33,40] did not report it.

Provider's perspective was adopted in twelve studies [17,20,22,24,28–30,32,34,37,38] whereas five studies [23,25,26,33,45] reported the societal perspective. Few studies employed other type of perspective which are mentioned in S1 Table.

Twenty-four studies used only direct cost for economic evaluation of TKR which included the cost of hospitalization and medication. Seven studies ([5,18,20,25,26,31,45] used both direct and indirect costs. Direct medical cost includes cost of medicine, diagnostic services, cost of implant, doctor's fee, and other hospitalization related expenditure and non-medical cost such as cost of food and cost of travel related to hospital access. Indirect cost includes income loss due to TKR and income loss of caretaker of the patient. One study did not report the type of cost used in cost-effectiveness analysis of TKR.

## Model structure and outcome

Eighteen [5–8,18,23,26–28,31–33,35,36,41,43,45] out of thirty-two studies used analytical model to report the outcomes like cost-effectiveness or cost-utility of TKR as mentioned in S1 Table. Twenty-five studies did not provide any justification regarding the scope of the model whereas Ferket et al [27] used marginal structure models for repeated measures and justified by stating that "outcome values can vary over time and can predict future treatment assignment along with other time-varying confounders". Similarly, Higashi et al [31] justified the use of discrete event simulation (DES) model to be a better approach than Markov model since larger number of attributes and events are more likely to be managed by the DES model as opposed to the Markov model. In addition to this, it is possible to attach memories in previous states in a DES model, which is difficult to achieve in a Markov model. Chen et al [41] used Osteoarthritis Policy Model (OAPol), a state-transition Monte Carlo computer simulation model to evaluate the epidemiologic factors that affect OA knee which calculates time, cost, and QALY gain while adjusting comorbidities. Dakin et al [35] used generalised linear model with gamma family distribution, as the costs and QALYs were highly skewed whereas Karmarkar et al [5] suggested that the objectives of the study could have been achieved through generalised linear model but the study used Markov model for its uniqueness to forecast the societal cost of disparities (race, ethnicity and sex) as Markov model was widely used to determine cost-effectiveness of single treatment modality.

Clear and transparent modelling methodologies were presented in all the included thirty-two studies. The source of data used to calculate transition probabilities were indicated in nine studies that used Markov model. Among these, seven included studies [5,7,23,26,28,36,45] incorporated transition probabilities from literature whereas only one study [18] used secondary data to calculate transition probabilities.

## Costing data

Out of the twenty-six studies selected for Purchasing Power Parity (PPP) analysis, nine studies were from USA [5,23–27,33,41,45], eight from European countries—UK [8,34,35], Denmark [6], France [36], Germany [37], Romania [38] and Spain [39], three from Canada [28–30],

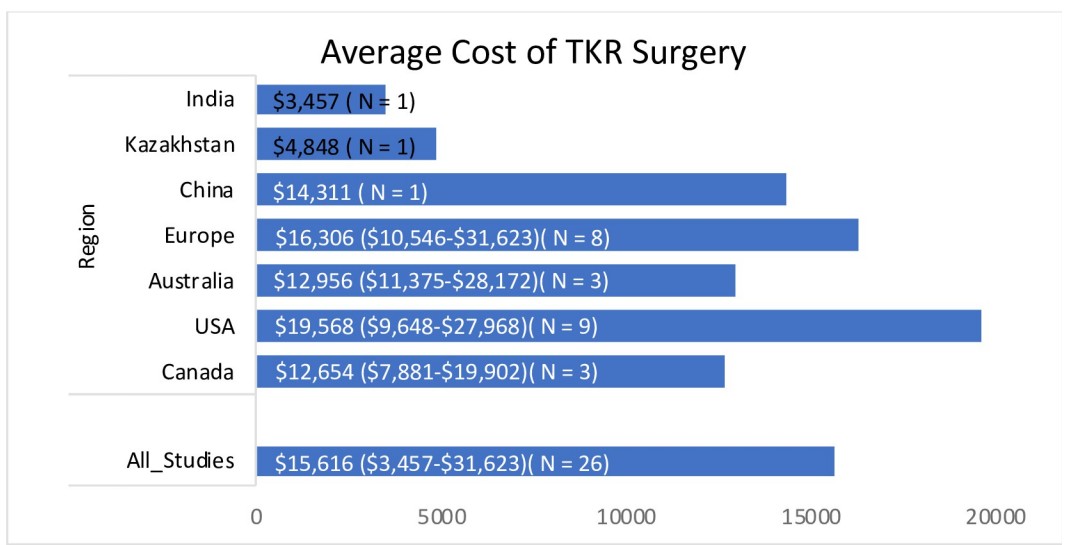

**Fig 2. Average cost of TKR surgery.**

three from Australia [31,32,43] one from India [7], one from China [40] and one from Kazakhstan [42]. Most studies [6,8,24,27,28,31,32,34,35,37–39] obtained the costing data from the national costing databases of respective countries whereas, few studies [5,7,29,40] used secondary data from available literature and few studies [23,25,26,33,45] used both national costing database and secondary data from literature.

The cost of TKR surgery—direct medical cost (Cost of hospitalization, medication, doctor's fee, surgery cost, and diagnostic cost)—in India was found to be the cheapest at $3457 followed by Kazakhstan at $4848 and China at $14,311. The average TKR cost in PPP ($) term was $15616 in 2021. It is lowest in developing country like India ($3457) and highest in USA ($19568). Region wise average cost of TKR surgery is presented in a bar graph as shown below in Fig 2.

## Quality of reporting

CHEERs checklist was used to evaluate the reporting quality of the studies. Twenty-three studies [5–7,17,18,20,23,25–28,30–32,35–38,40–43,45] were ranked as high, seven [8,19,22,29,33,34,39] as moderate and other two [21,24] were categorized as low. The proportion of studies that meet the requirements for each of the items has been compiled in S1 Fig. Only four studies directly stated the comparator name in the title, whereas majority of the studies went into great depth to describe their abstracts. Most of the studies reported information about setting, analytical model and assumptions, and discount rates. Additionally, eleven studies did not report the results of uncertainty analysis and nine studies did not report conflict of interest.

QHES was used to evaluate the quality of the studies included for the review. Twenty-one studies [5,6,17,18,22,23,25,27–31,35–39,41–43,45] were found to be of high quality and Nine [8,19,20,22,29,32–34,40] with fair quality and two studies [21,24] as poor quality. The discrepancies in scores were resolved through consensus by reviewers after discussion. Most of the studies explained the objectives in detail with the exception of Schilling et al [32]. Only one study [6] used RCT for the data analysis whereas, most studies used the primary data from hospital registries. Six studies used sub-group analysis where majority of studies used sensitivity

or uncertainty analysis. Nineteen studies used economic analysis to evaluate the cost effectiveness of TKR. The detailed scoring of QHES of all studies are given in S2 Fig.

## Discussion

TKR is a widely used clinical intervention for the treatment of knee osteoarthritis [46–48]. It is also considered to be effective among all age groups, especially among septuagenarians and octogenarians [49,50]. Our literature review suggests that TKR surgery is often considered a cost-effective treatment method for OA knee compared to the non-surgical management, primarily based on studies conducted in the developed countries [23,28,45]. Given the increasing use of TKR due to its clinical effectiveness, this review generates crucial evidence on core modelling specifications on cost-effectiveness of TKR compared to the non-surgical management. Further, this also presented cost of different countries in a single currency for international comparison by converting different countries cost to international dollar ($) using purchasing power parity (PPP) method.

This systematic review assessed the models used in cost utility or cost-effectiveness of the thirty-two studies, out of which, eighteen studies reported methodology to determine cost-effectiveness. Nine studies used Markov model [5,7,18,23,26,28,36,43,45] whereas five studies reported regression model [6,32,33,35,37]. One study each reported Marginal structure model [27], discrete simulation model [31], decision tree analysis [8], and Osteoarthritis Policy Model (OAPol) [41] to assess the cost-effectiveness of TKR.

By comparing different models used in our literature review which, included high quality studies as per CHEERS and QHES checklist, our findings suggested that the Markov model was most widely used in the analysis of the cost effectiveness of TKR. This is similar to the results of previous economic studies [7,28,36] which, showed that Markov model allows a better comparative understanding of the cost and health outcome across interventions. According to Kuntz et al. "the primary difference between a Markov model and a decision tree is that the former models the risk of recurrent events over time in a straightforward fashion" [51]. The Markov model can also be used in different time frames and study settings.

In the context of TKR surgery, where the economic evaluation of outcomes is closely tied to quality-adjusted life years (QALY), especially in cases involving physical disability, the Markov model provides a more straightforward interpretation of the interconnected stages and sub-stages specific to TKR compared to alternative models. Moreover, in the case of TKR, there is likelihood of the patients transitioning from one state to other and again reversing and as the Markov model captures the recurrent events, this is preferred over other model for analysing the dynamic events. It is also found that the Markov model works better with small datasets and it is convenient for TKR [18,37,40]. It also incorporates both actual data and hypothetical data based on assumptions, enabling the calculation of the incremental cost-effectiveness ratio (ICER). Moreover, actual data can also be utilized to assess the budget impact of the intervention. Therefore, the Markov model not only allows for the calculation of the country-level effect and cost of new interventions but also provides insights into the economic implications of implementing such interventions.

Our review also summarises that most of the studies were of high quality as per the CHEERs and QHES checklist. Twenty-three studies were defined as high quality as per the CHEERs checklist and twenty-one as per the QHES, indicating a higher percentage of studies, met the quality standard.

Of the thirty-two studies in our systematic review, thirty-one studies used direct cost to analyse the precise cost of the intervention. Although indirect cost was used in seven studies

which explained the total economic burden of TKR, but could not be used for PPP as the economic burden includes opportunity cost of time which is difficult to express in PPP terms.

Out of thirty-two studies, three studies were excluded because of older than 2000s and another three studies did not report the actual cost, hence direct cost of TKR from twenty-six studies was used for costing analysis using PPP. The direct cost of TKR by each study was reported in their respective country currency. We converted the cost of TKR reported in all included studies to international dollar ($) through PPP method in order to compare across different countries.

The cost of TKR surgery widely varied across different countries. Another important finding of our review is that the cost of TKR surgery in a single currency (international $) was found to be lowest in India and highest in USA. It was also found that the cost of TKR surgery was three to five times higher in the developed countries—USA and UK than the developing countries—India. These variations in cost might be due to the difference in the cost of human resources. Despite the high cost of TKR in the developed countries, the TKR surgery was found to be cost effective than non-surgical management for OA knee [45]. Hence, TKR is also found to be cost-effective in developing countries such as India.

## Strengths

Most of the systematic review on TKR surgery summarised the degree of cost-effectiveness [10–12] and one study presented the study design [13]. Our review is the first study to suggest a suitable model to estimate the cost effectiveness especially for TKR compared to non-surgical management. Based on our study findings, new studies can utilize Markov model for their analysis and accordingly focus on the appropriate data requirements. Another strength of this review is that our study is the first study to use PPP to convert the cost of TKR surgery into a single currency to compare the cost of TKR across different countries.

Our review used both CHEERS and QHES to measure the quality of studies. While CHEERS measured the overall quality of the studies, QHES measured the quality of economic evaluation used in the studies. Overall, most studies scored well on both the scales and are of high quality.

## Limitations

Since studies included in our review were from various countries, the generalisability of our findings may be reduced due to the variations in costs and clinical practices across multiple continents. Most of the included studies were from the high-income countries with an exception of two studies from the low and middle-income countries. Therefore, application research findings to the low and middle-income countries might be a problem. However, the PPP method used to convert currency and appropriate methods as identified by this review may provide insights for uptake of these findings in other country settings.

## Conclusion

The review provides critical insights for guiding model specification for conducting cost effectiveness analysis of TKR as one of the clinical interventions compared to the non-surgical management. This review also underlines the merits of Markov model in economic evaluation studies related to TKR. It is also found that the overall quality of reporting in the cost effectiveness studies is increasing globally however, there are limited number of studies in the low and middle income countries. This suggests for undertaking more CEA studies in these countries as the burden of OA knee is increasing. This review concluded that Markov model is the most

suitable decision model for economic evaluation of TKR and non-surgical management. More studies with high methodological standards in developing countries are recommended.

## Supporting information

**S1 Checklist. Prisma 2020 checklist.**
(DOCX)

**S1 Fig. CHEERS checklist.**
(TIF)

**S2 Fig. QHES.**
(TIF)

**S1 Table. Models and cost of TKR used across studies.**
(DOCX)

**S1 Appendix. Search strategies.**
(DOCX)

## Author Contributions

**Conceptualization:** Naline Gandhi, Amatullah Sana Qadeer, Sarit Kumar Rout.

**Data curation:** Naline Gandhi, Amatullah Sana Qadeer, Abhilash Patra, Jebamalar John, Aiswarya Anilkumar.

**Formal analysis:** Naline Gandhi, Amatullah Sana Qadeer, Ananda Meher, Jebamalar John, Aiswarya Anilkumar.

**Funding acquisition:** Lipika Nanda.

**Investigation:** Naline Gandhi, Amatullah Sana Qadeer, Ananda Meher, Jebamalar John, Aiswarya Anilkumar.

**Methodology:** Naline Gandhi, Amatullah Sana Qadeer, Ananda Meher, Jennifer Rachel, Abhilash Patra, Sarit Kumar Rout.

**Project administration:** Amatullah Sana Qadeer.

**Resources:** Lipika Nanda.

**Supervision:** Ambarish Dutta, Lipika Nanda, Sarit Kumar Rout.

**Validation:** Naline Gandhi, Ambarish Dutta, Lipika Nanda, Sarit Kumar Rout.

**Visualization:** Naline Gandhi, Amatullah Sana Qadeer.

**Writing – original draft:** Naline Gandhi, Amatullah Sana Qadeer, Jebamalar John, Aiswarya Anilkumar.

**Writing – review & editing:** Amatullah Sana Qadeer, Ananda Meher, Jennifer Rachel, Abhilash Patra, Sarit Kumar Rout.

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
