## [Decision Letter · Decision Letter 0]

22 May 2023

PONE-D-22-35337Costs and models used in the economic analysis of Total Knee Replacement (TKR): A Systematic ReviewPLOS ONE

Dear Dr. Rout,

Thank you for submitting your manuscript to PLOS ONE. After careful consideration, we feel that it has merit but does not fully meet PLOS ONE’s publication criteria as it currently stands. Therefore, we invite you to submit a revised version of the manuscript that addresses the points raised during the review process.

We look forward to receiving your revised manuscript.

Kind regards,

Meng Li

Academic Editor

PLOS ONE

3. We note that the manuscript is reporting a meta-analysis on genetic association studies. We need you to provide us with additional information in relation to this meta-analysis; please complete the following checklist and upload it as a Supporting Information file with a file name “GAMA checklist”. The checklist can be downloaded here: http://www.plos.org/wp-content/uploads/2013/05/meta-analysis-on-genetic-association-studies-form.docx.

Reviewers' comments:

Reviewer's Responses to Questions

**Comments to the Author**

1. Is the manuscript technically sound, and do the data support the conclusions?

Reviewer #1: Yes

Reviewer #2: Yes

2. Has the statistical analysis been performed appropriately and rigorously? 

Reviewer #1: Yes

Reviewer #2: Yes

3. Have the authors made all data underlying the findings in their manuscript fully available?

Reviewer #1: Yes

Reviewer #2: Yes

4. Is the manuscript presented in an intelligible fashion and written in standard English?

Reviewer #1: Yes

Reviewer #2: Yes

5. Review Comments to the Author

Reviewer #1: - The study on ‘costs and models used in the economic analysis of Total Knee Replacement (TKR) – a systematic review’ is simple, clear and to the point. It is useful in devising best model for cost effectiveness that can be utilized in further studies. The review also highlights the point that low- and middle-income countries despite having limited number of studies also have lowest cost of TKR surgery. The cost extraction through PPP method is helpful in identifying the best medical tourism for TKR across different countries of the world.

- References 7 and 9 are the same, repeated.

Reviewer #2: 1. The study overall is well-structured and clearly written.

2. Google Scholar was not mentioned as a resource for literature in the methodology and abstract sections but appeared in the results section.

3. The discussions regarding the Markov model (Lines 334-342) were general and focused on the characteristics of the model. It would have been beneficial to have a more comprehensive discussion, specifically addressing why the Markov model is more applicable or appropriate for TKR.

4. The definition of direct and indirect costs needs clarification. For example, in Line 257, the authors mentioned "cost of travel." If this refers to transportation costs covered by health insurance, it might fall under direct non-medical costs. It would be helpful to explain this in more detail, perhaps by providing a brief explanation in the text or adding footnotes to Table 1.

5. The section pertaining to the cost data for TKR surgery is somewhat ambiguous (Line 288-296). Could you please clarify the reporting process for the cost of TKR surgery in this section? It's unclear if ancillary costs were factored into the reported cost or if it is strictly focused on the surgical expenses. Also, does the reported cost represent the price per surgery? Your clarification would greatly enhance our understanding of the data.

6. In Line 320-322, you mentioned that "The TKR surgery is also a cost-effective method of treatment for OA knee compared to non-surgical management across the world, which mostly consists of studies from developed countries." This statement might be misleading to readers. It is important to clarify that this statement reflects findings from existing research rather than a conclusion drawn from your own study. Consider revising it to: "Previous literature reviews indicated that TKR surgery is often considered a cost-effective treatment method for OA knee compared to non-surgical management, primarily based on studies conducted in developed countries worldwide."

7. Needs some adjustments and format change for improved consistency, clarity and readability

(1) Figure 1, particularly the "Reports not retrieved" section is incomplete.

(2) Several typo errors and instances of missing punctuation have been noticed, such as in Line 324. Kindly make the necessary corrections.

(3) For Figure 2, it is recommended to use the currency format ($##,###).

8. Did you consult with a librarian to develop your search terms? If so, please mention in your methods.

6. PLOS authors have the option to publish the peer review history of their article (what does this mean?). If published, this will include your full peer review and any attached files.

Reviewer #1: **Yes: **Dr. Madhuri Latha Devaraju

Reviewer #2: No

---

## [Author Response · Author response to Decision Letter 0]

26 Jun 2023

Dear Editor,

 I, on behalf of all the authors would like to thank you for your valuable comments and suggestions, which gave us the opportunity to improve the paper. In the revised manuscript, we have addressed all the issues raised. Moreover, in this document, I provide point by point response to queries raised by the journal. Comments are shown in bold font, followed by our answer/comment in normal font. The major corrections/changes in the manuscript are displayed in red font.

We have revised the manuscript to meet PLOS ONE’s style requirements, including file naming according to the templates provided

We have clarified in detail, the fund receipt process for developing the manuscript. Department of health research (DHR), MoHFW, Government of India, has signed a MoU with IIPH Hyderabad to establish a regional resource center for conducting Health technology assessment (HTA). This resource center as a part of its ongoing research conducted this literature review. The staff engaged in this activity were funded by the DHR.

3. We note that the manuscript is reporting a meta-analysis on genetic association studies. We need you to provide us with additional information in relation to this meta-analysis; please complete the following checklist and upload it as a Supporting Information file with a file name “GAMA checklist”. The checklist can be downloaded here: http://www.plos.org/wp-content/uploads/2013/05/meta-analysis-on-genetic-association-studies-form.docx.

The manuscript does not have a meta-analysis component, nor does it report on genetic studies. Hence, a GAMA checklist is not applicable to our study.

Reviewer #1: 

The study on ‘costs and models used in the economic analysis of Total Knee Replacement (TKR) – a systematic review’ is simple, clear and to the point. It is useful in devising best model for cost effectiveness that can be utilized in further studies. The review also highlights the point that low- and middle-income countries despite having limited number of studies also have lowest cost of TKR surgery. The cost extraction through PPP method is helpful in identifying the best medical tourism for TKR across different countries of the world.

1. References 7 and 9 are the same, repeated.

Thank you very much for your detailed observation and we have made the necessary changes in the manuscript. 

Reviewer #2: 

1. The study overall is well-structured and clearly written.

2. Google Scholar was not mentioned as a resource for literature in the methodology and abstract sections but appeared in the results section.

We have changed as per the suggestions in all the sections. 

3. The discussions regarding the Markov model (Lines 334-342) were general and focused on the characteristics of the model. It would have been beneficial to have a more comprehensive discussion, specifically addressing why the Markov model is more applicable or appropriate for TKR.

This is an important observation and in the revised manuscript, we have explicitly mentioned the merits of Markov model as this captures different health states when a person can move from one state to other and again reverses. When TKR surgery is done, the patient may transition from bad to good and further may return to a worse situation, Markov Model captures all these transitions and provides their ICER values. 

We have elaborated on this in Lines 353-371.

4. The definition of direct and indirect costs needs clarification. For example, in Line 257, the authors mentioned "cost of travel." If this refers to transportation costs covered by health insurance, it might fall under direct non-medical costs. It would be helpful to explain this in more detail, perhaps by providing a brief explanation in the text or adding footnotes to Table 1.

We have clarified the definition of direct and indirect cost in Lines (261-264) as used by the manuscript under review. Importantly, direct cost includes cost of medicine, diagnostic services, cost of implant, doctor’s fee and other hospitalization related expenditure, and non-medical cost such as cost of food, travel related to hospital access. Indirect cost includes —income loss due to TKR and income loss of caretaker of the patient.

5. The section pertaining to the cost data for TKR surgery is somewhat ambiguous (Line 288-296). Could you please clarify the reporting process for the cost of TKR surgery in this section? It's unclear if ancillary costs were factored into the reported cost or if it is strictly focused on the surgical expenses. Also, does the reported cost represent the price per surgery? Your clarification would greatly enhance our understanding of the data.

We have added details on data collection of costs in Lines (299-302). Moreover, studies that collected costs from national database and secondary literature are cited as well.

6. In Line 320-322, you mentioned that "The TKR surgery is also a cost-effective method of treatment for OA knee compared to non-surgical management across the world, which mostly consists of studies from developed countries." This statement might be misleading to readers. It is important to clarify that this statement reflects findings from existing research rather than a conclusion drawn from your own study. Consider revising it to: "Previous literature reviews indicated that TKR surgery is often considered a cost-effective treatment method for OA knee compared to non-surgical management, primarily based on studies conducted in developed countries worldwide."

We have revised it to “Our literature review suggests that TKR surgery is often considered a cost-effective treatment method for OA knee compared to the non-surgical management, primarily based on studies conducted in the developed countries”

7. Needs some adjustments and format change for improved consistency, clarity and readability

(1) Figure 1, particularly the "Reports not retrieved" section is incomplete.

(2) Several typo errors and instances of missing punctuation have been noticed, such as in Line 324. Kindly make the necessary corrections.

(3) For Figure 2, it is recommended to use the currency format ($##,###).

Thank you for the suggestions. The changes have been incorporated accordingly.

8. Did you consult with a librarian to develop your search terms? If so, please mention in your methods.

We did not consult a librarian during the development of search terms or at any point of the review.

---

## [Decision Letter · Decision Letter 1]

13 Jul 2023

Costs and models used in the economic analysis of Total Knee Replacement (TKR): A Systematic Review

PONE-D-22-35337R1

Dear Dr. Rout,

We’re pleased to inform you that your manuscript has been judged scientifically suitable for publication and will be formally accepted for publication once it meets all outstanding technical requirements.

Kind regards,

Meng Li

Academic Editor

PLOS ONE

Additional Editor Comments (optional):

Reviewers' comments:

Reviewer's Responses to Questions

**Comments to the Author**

1. If the authors have adequately addressed your comments raised in a previous round of review and you feel that this manuscript is now acceptable for publication, you may indicate that here to bypass the “Comments to the Author” section, enter your conflict of interest statement in the “Confidential to Editor” section, and submit your "Accept" recommendation.

Reviewer #1: All comments have been addressed

2. Is the manuscript technically sound, and do the data support the conclusions?

Reviewer #1: Yes

3. Has the statistical analysis been performed appropriately and rigorously? 

Reviewer #1: Yes

4. Have the authors made all data underlying the findings in their manuscript fully available?

Reviewer #1: Yes

5. Is the manuscript presented in an intelligible fashion and written in standard English?

Reviewer #1: Yes

6. Review Comments to the Author

Reviewer #1: (No Response)

7. PLOS authors have the option to publish the peer review history of their article (what does this mean?). If published, this will include your full peer review and any attached files.

Reviewer #1: **Yes: **Madhuri Devaraju

---

## [Editor Report · Acceptance letter]

17 Jul 2023

PONE-D-22-35337R1 

Costs and models used in the economic analysis of Total Knee Replacement (TKR): A Systematic Review 

Dear Dr. Rout:

I'm pleased to inform you that your manuscript has been deemed suitable for publication in PLOS ONE. Congratulations! Your manuscript is now with our production department. 

Kind regards, 

on behalf of

Dr. Meng Li 

Academic Editor

PLOS ONE